# Hybrid passive micromixer using combined traditional microfabrication and 3D printing for gold nanoparticle synthesis

Yasser Aldaghestani[1,2], Andreas Schiffer[1,3]*, Anas Alazzam [1,2]*

1 Department of Mechanical and Nuclear Engineering, Khalifa University of Science and Technology, Abu Dhabi, United Arab Emirates, 2 System on Chip Lab, Khalifa University of Science and Technology, Abu Dhabi, United Arab Emirates, 3 Advanced Research and Innovation Center (ARIC), Khalifa University of Science and Technology, Abu Dhabi, United Arab Emirates

* andreas.schiffer@ku.ac.ae (AS); anas.alazzam@ku.ac.ae (AA)

## Abstract

This study introduces a novel hybrid passive micromixer that seamlessly integrates conventional microfabrication techniques with cost-effective 3D printing to facilitate the controlled synthesis of nanoparticles with high precision in size and morphology. The micromixer design combines a Y-junction microchannel fabricated using soft lithography with an embedded 3D-printed helical structure to enhance mixing efficiency. Notably, the helical structure was fabricated using a commercially available, cost-effective Digital Light Processing (DLP) 3D printer, demonstrating that high-performance microfluidic devices can be manufactured without reliance on expensive, specialized equipment. Numerical simulations were conducted to analyze the micromixer parametrically and to compare its performance to various micromixer designs reported in the literature. The results reveal the superiority of the mixing capabilities of the proposed mixer under laminar flow conditions. Experimental validation using dyed fluids and image analysis techniques confirmed the enhanced mixing performance of the hybrid micromixer, reaching a mixing efficiency of 92% at a Reynolds number of 1, with results closely matching the numerical predictions. The device was then used in the synthesis of gold nanoparticles, employing L-ascorbic acid as the reducing agent. Characterization of the synthesized nanoparticles via UV-Vis spectroscopy and scanning electron microscopy (SEM) demonstrated precise control over particle size and distribution, with gold nanoparticles ranging from 14 to 25 nm at a total flow rate of 5000 µL/min. This hybrid micromixer approach offers a scalable, efficient, and accessible platform for nanomaterial synthesis, with potential applications in fields such as drug delivery, biosensing, and catalysis.

**Data availability statement:** All relevant data are within the paper and its Supporting Information files.

**Funding:** The author(s) received no specific funding for this work.

**Competing interests:** The authors have declared that no competing interests exist.

## Introduction

Microfluidic devices have emerged as powerful tools for the precise manipulation and control of fluids at the micro- and nanoscale, enabling complex operations to be integrated into compact lab-on-a-chip platforms [1,2]. These systems have been widely applied in areas such as nanomaterial synthesis [3], biomedical diagnostics [4], drug discovery [5], and cell separation and analysis [6,7]. The miniaturization of fluidic processes offers several advantages, including reduced reagent consumption, enhanced heat and mass transfer, improved reaction control, and the ability to produce uniform products with narrow size distributions [8–10].

Efficient mixing is a critical requirement in many microfluidic applications, particularly for chemical reactions and nanoparticle synthesis [11,12]. However, due to the laminar flow regime at low Reynolds numbers, mixing in microchannels is typically limited to slow molecular diffusion [13]. To overcome this limitation, a wide range of micromixer designs have been developed and are commonly classified as active or passive. Active micromixers employ external energy sources, such as acoustic [14–16], magnetic [17], or electric fields [18], to enhance mixing, whereas passive micromixers rely on channel geometries and surface features to induce chaotic advection and fluid interface stretching [19,20]. Although active mixers can achieve high mixing efficiencies, passive micromixers are often preferred due to their simplicity, ease of integration, and lack of external actuation requirements.

Passive micromixers have been widely employed for nanoparticle synthesis, including lamination-based designs, hydrodynamic flow-focusing devices, and Y- and T-junction configurations [21–25]. Microfluidic synthesis of nanoparticles offers improved control over particle size, morphology, and uniformity compared to batch processes, making it particularly attractive for producing nanoparticles with high-quality physicochemical properties [26]. However, several practical challenges remain. In particular, insufficient mixing and nanoparticle adhesion to microchannel walls can lead to particle heterogeneity, fouling, and channel clogging during continuous operation [27,28].

To overcome these limitations, 3D printing has emerged as a promising alternative method for fabricating microfluidic devices [29]. It enables easy and fast iterative prototyping and manufacturing, making it ideal for testing-based optimization. Additionally, 3D-printed devices do not require a cleanroom fabrication environment, reducing costs. Additive manufacturing (AM) technologies have played a pivotal role in advancing the fabrication of intricate and precise devices. Among these technologies, Material Extrusion [30], Vat Photopolymerization [31], and Material Jetting [32] stand out for their significant contributions.

Spiral and helical microchannel architectures provide an effective passive strategy for enhancing micromixing by exploiting curvature-induced Dean vortices, which generate strong transverse convection even at low Reynolds numbers [33,34]. In the context of continuous nanoparticle synthesis, enhanced micromixing shortens residence-time distributions and accelerates nucleation and early growth processes, resulting in more uniform particle sizes and improved reproducibility compared to

straight-channel designs [35,36]. These advantages have motivated the use of spiral and helical geometries as robust passive micromixers for high-throughput nanoparticle production.

Building on these principles, Erfle et al. [37,38] employed two-photon polymerization (2PP) to fabricate advanced 3D microfluidic micromixers for lipid nanoparticle (LNP) synthesis, enabling precise control over complex mixing geometries that cannot be realized using conventional planar microfabrication. Their Coaxial Lamination Mixer (CLM) integrates coaxial injection, stretch-and-fold elements, and inlet filtering to significantly suppress fouling during nanoparticle synthesis [37]. Subsequently, the Horseshoe Lamination Mixer (HLM) achieved rapid and homogeneous mixing, producing monodisperse LNPs with particle sizes ranging from 42 to 166 nm and low polydispersity indices [38]. These studies demonstrate the potential of 3D printed microstructures/microchannels to enhance micromixing and nanoparticle synthesis performance; however, the need for specialized, high-cost 2PP fabrication restricts scalability and limits widespread adoption.

In this work, the development and fabrication of a hybrid passive micromixer are presented. It is termed "hybrid" because it integrates traditional microfabrication methods with cost-effective 3D printing and is designed for efficient gold nanoparticle synthesis. The microchannel was fabricated using soft lithography, while a 3D-printed helical structure was embedded within the channel to enhance mixing efficiency. Unlike previous studies that rely on costly 2PP 3D printers for high-resolution microstructures, the current design utilizes a cost-efficient DLP printer. This approach significantly reduces fabrication costs, making advanced microfluidic devices accessible to a broader range of researchers. The proposed micromixer addresses limitations associated with passive micromixers by incorporating a helical structure, facilitating the production of uniform and scalable nanoparticles. Numerical simulations were conducted to assess the efficiency of the hybrid passive micromixer, and experimental validation was performed using dye-based mixing studies.

## Materials and methods

This section describes the methodology employed to design, fabricate, and characterize a hybrid passive micromixer incorporating a 3D-printed helical structure for enhanced fluid mixing. Detailed descriptions of the materials, fabrication protocols for the helical structure and microchannel, and experimental procedures for evaluating mixing efficiency and synthesizing gold nanoparticles are provided. Additionally, nanoparticle synthesis, purification, and characterization protocols, utilizing scanning electron microscopy (SEM) and UV-Vis spectroscopy, are outlined.

### Materials

The following materials were utilized in the experiments: Silicon wafer (4-inch diameter, 500 µm thick), 3D printing Resin (JAMG HE Standard Plus Resin Grey), SU-8 2150 negative photoresist (MicroChem Corp, Newton, US), Polydimethyl-siloxane (PDMS) (Sylgard 184, Dow Corning, Michigan, US), isopropanol (IPA) (absolute, > 99.8%, Sigma-Aldrich, USA), ethanol (absolute, > 99.8%, Sigma-Aldrich, USA), gold(III) chloride trihydrate (Sigma-Aldrich, purity ≥ 99.9%), L-ascorbic acid (Sigma-Aldrich), and deionized water (DI).

### 3D-printed helical structure

The fabrication of the helical structure, a key component in enhancing mixing efficiency within the microchannel, was achieved through DLP 3D printing using the ELEGOO Saturn 3 Ultra printer, which features a 19 x 24 µm XY resolution and a 405 nm UV light source. The chosen material for printing is JAMG HE Standard Plus Resin Grey, which is selected for its durability and precision in capturing complex geometries. The printing parameters were set according to the manufacturer's recommendations, with a slice thickness of 0.01 mm and a bottom layer count of 7. The initial layer's curing time was 40 s, and subsequent layers were cured for 6 s each, ensuring optimal solidity and dimensional accuracy. After printing, the structure was rinsed with ethanol to remove any excess resin and then subjected to UV post-curing to achieve the desired mechanical properties. The helix was designed using SolidWorks software, with a total length of 20 mm and an outer diameter of 0.5 mm. It had a round cross-section with a diameter of 0.2 mm and featured approximately 20

revolutions. The precise dimensions, designed in SolidWorks, are critical for ensuring an optimal fit within the microchannel, thereby maintaining the secure placement of the helical structure.

Fabrication of the entire microfluidic device by DLP alone was deemed impractical due to the difficulty of removing support material from enclosed channels, as well as the fact that all-printed channels exhibit higher surface roughness, dimensional variability in sub-100-μm gaps, and reduced transparency.

## Microchannel design and fabrication

As depicted in Fig 1, the fabrication process of the hybrid microchannel incorporates a 3D element integrated within its structure. The microchannel consists of two components: a Polydimethylsiloxane (PDMS) channel, fabricated using standard soft lithography, and a DLP-printed helical structure made from a commercial photopolymer resin, which is an acrylate-based UV-curable resin that forms a crosslinked polymer network upon photopolymerization. The PDMS channel is created through conventional soft lithography techniques, which involve forming a replica mold on a silicon wafer using SU8–2150 photoresist. The SU8 layer, with a thickness of 500 μm, is fabricated using a direct lithography system (Dilase 650, KLOE), ensuring precise patterning and high aspect ratio capabilities.

Before bonding the PDMS channel to the substrate, the 3D-printed helical structure was inserted into the channel using a press-fit assembly. The square channel cross-section (500 μm × 500 μm) was designed to exactly match the 500 μm outer diameter of the helical structure, resulting in a perfect dimensional match that enabled self-centering and stable alignment along the entire channel length. Additionally, the microchannel length was designed to be slightly longer than the helical structure to facilitate ease of insertion and handling during assembly. Therefore, the structure could be placed without the need for an external alignment system. The helical insert was manually positioned using tweezers, and its placement was visually verified under an optical microscope prior to sealing. To prevent axial migration of the helical structure during operation, a downstream constriction was incorporated at the end of the microchannel (Fig 2b). The

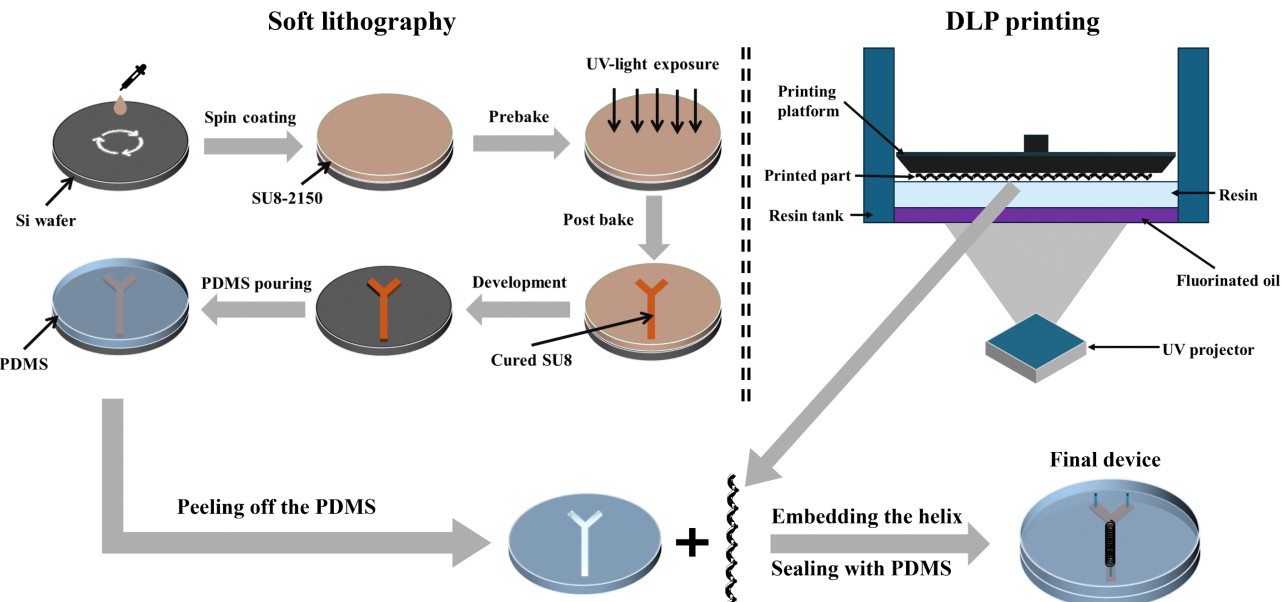

**Fig 1. Schematic illustration of the hybrid microchannel fabrication process.** The PDMS microchannel is produced by soft lithography, and the helical structure is fabricated by DLP 3D printing. The two parts are combined and bonded to a PDMS substrate to form the final device.

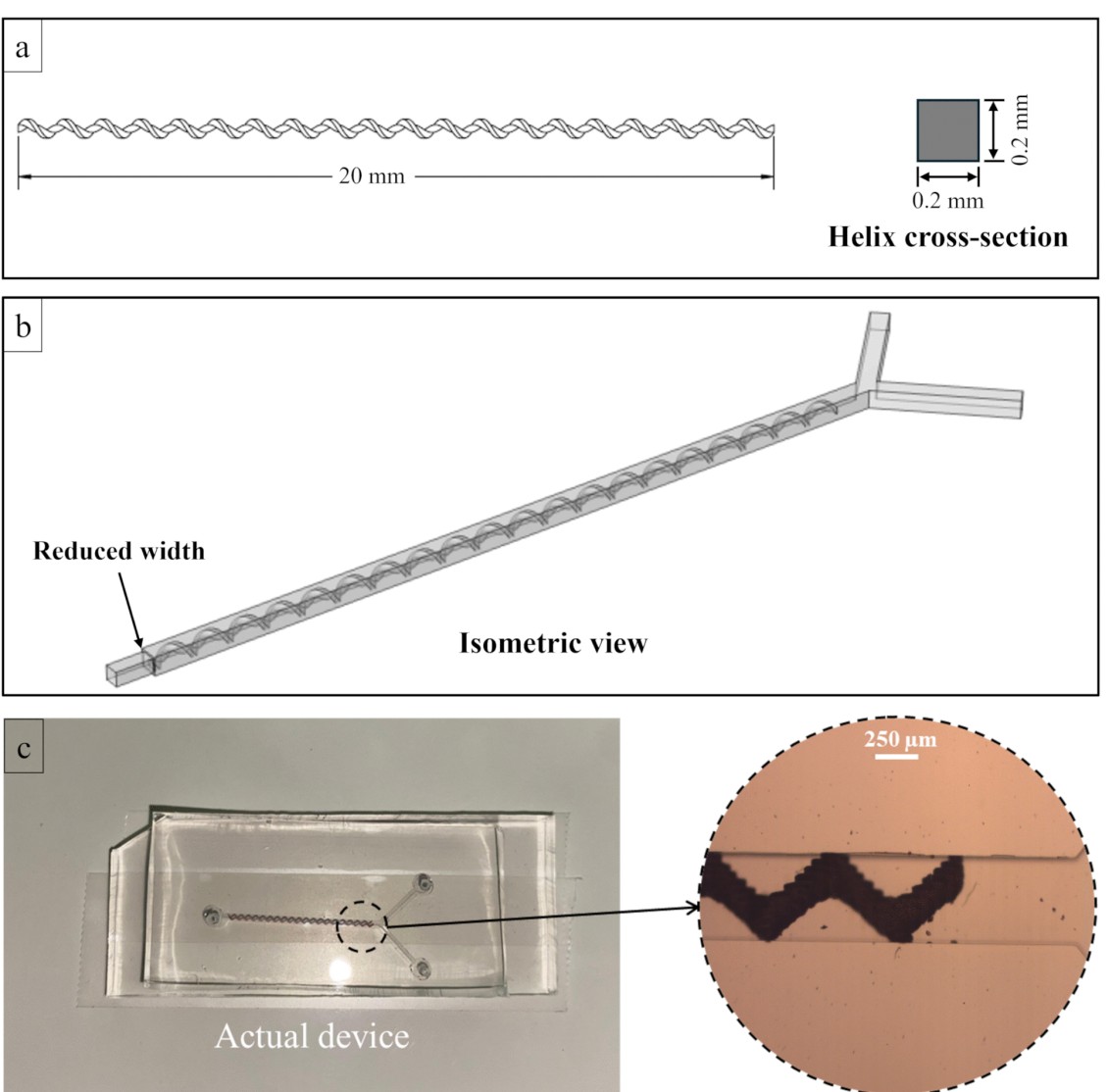

**Fig 2. Design and integration of the 3D-printed helical structure within the hybrid passive micromixer. (a)** Schematic illustration of the helical 3D-printed part, showing overall geometry and key dimensions, with an inset depicting the channel cross-section. **(b)** Schematic isometric view of the helical structure **(c)** Photograph of the 3D-printed element inserted into the microchannel, with an inset showing a microscopic image of the helical structure within the channel, demonstrating proper alignment and fit.

channel width was locally reduced by 50 µm on each side, forming a 400 µm-wide restriction that is smaller than the outer diameter of the helical structure. This constriction provided stable axial retention of the helical structure at total flow rates up to 5000 µL/min, with no observable displacement during nanoparticle synthesis. No bonding between the structure and PDMS is required; the structure is physically captured by the channel walls/stop and subsequently encapsulated when the microchannel is oxygen-plasma bonded to the substrate. No surface treatment was applied to the 3D-printed structure; oxygen plasma was used exclusively to activate PDMS-PDMS bonding. Across the experiments reported here, no leakage or delamination of the encapsulated structure was observed. The design of the 3D-printed helical structure and the overall architecture of the hybrid passive micromixer are shown in Fig 2.

A hybrid fabrication approach combining soft lithography and DLP 3D printing was employed to exploit the complementary advantages of conventional microfabrication and additive manufacturing. Soft lithography was used to produce the straight microchannel, providing long, optically transparent channels with smooth walls and high dimensional fidelity, which facilitate imaging and inspection and enable robust, permanent bonding to the substrate. This approach minimizes fouling, delamination, and leakage, ensuring reliable operation under the applied flow conditions. The helical structure was fabricated via DLP 3D printing, allowing realization of its fully three-dimensional geometry, which is not achievable using conventional planar, mold-based techniques. DLP was preferred over 2PP due to its lower cost, higher throughput, and wider accessibility, while still providing sufficient resolution for this work. Typical DLP systems achieve feature resolutions of 25–50 μm, which meet the resolution requirements for the helical structure and channel geometry used in this study. Although 2PP offers much higher resolution (100–200 nm), such sub-micron precision is unnecessary for the feature sizes in this design and would significantly increase fabrication time and cost.

## Microfluidic experiments

A series of experiments were conducted to evaluate the functionality of the fabricated microfluidic device. These experiments were designed to address two key aspects: first, to quantify the mixing performance within the microchannel, and second, to demonstrate the device's capability for controlled nanoparticle synthesis.

## Mixing characterization

Two fluids of distinct colors were injected into the hybrid microfluidic system to assess the mixing performance. These dyed water solutions were employed to visually quantify the mixing process within the channels. The experiments were conducted at flow rates of 30, 111, and 278 μL/min, corresponding to Reynolds numbers of 1, 3.7, and 9.27, respectively. The fluids were pumped through the microchannel using a NeMESYS syringe pump (Cetoni GmbH) equipped with 1 mL glass syringes. The microchannel was securely positioned on the stage of a Zeiss Axio Observer inverted microscope, with transmitted light utilized for observation. A high-speed Photron Fastcam SA-X2 camera, integrated with the microscope, was employed to capture the images.

Image analysis was performed in MATLAB to estimate the mixing performance within the microchannel. The captured images were first converted from the RGB color space to the CIELAB color space, allowing for more precise color analysis. The red channel intensity from designated lines across the image was then analyzed, with the number of lines determined by user input. The red channel was chosen because it provided the highest contrast and most reliable intensity variations for assessing mixing performance in the microchannel. This ensured better sensitivity, reduced noise from other channels, and simplified data processing while still accurately capturing mixing efficiency. Using MATLAB's "ginput" function, points along the microchannel were selected, and the "improfile" function was used to extract intensity profiles along these lines. The extracted profiles were normalized against the initial profile (at the inlet), and variances were calculated as described in the methodology. Lower variances indicate more uniform mixing, making this technique a direct and effective tool for assessing the efficiency of microchannel mixing.

## Nanoparticle synthesis

Gold nanoparticles are conventionally synthesized using macroscopic stirred vials or batch reactors, where particle size is mainly controlled by adjusting precursor concentration, reducing-agent stoichiometry, temperature, or reaction time. In such systems, mixing is achieved by bulk stirring. Since nanoparticle formation generally occurs in environments with millimeter- or centimeter-scale dimensions, local fluctuations in precursor concentration can develop, resulting in particle size heterogeneity and variation [39]. For example, Ekaputra et al. [40] reported that gold nanoparticles synthesized in stirred batch reactors using $HAuCl_4$ and L-ascorbic acid exhibited average diameters of approximately 20–40 nm at

alkaline pH (pH 10–11), while larger particles of 200–300 nm were obtained under acidic conditions (pH 3–4), demonstrating that particle size in macroscopic systems is governed by solution chemistry rather than hydrodynamic control. By contrast, microfluidic synthesis operates under laminar flow conditions, where mixing and residence time are strongly influenced by channel geometry and flow rate. As a result, in microfluidic systems, particle formation can be modulated by adjusting flow-rate-controlled residence time while maintaining identical reactant concentrations, offering a different parameter space for controlling nanoparticle synthesis compared to conventional batch approaches.

The hybrid microchannel was used to synthesize gold nanoparticles by mixing gold precursor and reducing agent. The gold precursor was prepared by dissolving an accurately weighed amount of chloroauric acid ($HAuCl_4$) in deionized water to obtain a one mM solution. The solution was stirred at an ambient temperature until complete dissolution was achieved, resulting in a homogeneous mixture. In a separate procedure, a one mM solution of L-ascorbic acid was freshly prepared by dissolving a measured quantity of reagent-grade L-ascorbic acid in deionized water, followed by gentle stirring and brief sonication to ensure complete dissolution and the elimination of entrapped air bubbles. Both solutions were subsequently injected into a hybrid micromixer to facilitate the synthesis of gold nanoparticles. In this process, the mixing induced the reduction of $Au^{3+}$ to $Au^0$, a transformation that was visually evident through the characteristic color change.

The fabrication of gold nanoparticles was investigated at varying flow rates using the same hybrid micromixer. After each run, a thorough washing procedure was implemented to remove residual synthesis reagents and contaminants. All nanoparticle samples underwent a rigorous washing procedure designed to optimize their suitability for scanning electron microscopy (SEM) analysis and ultraviolet-visible (UV-Vis) spectroscopy. This purification process effectively eliminates residual synthesis reagents, contaminants, and interfering by-products, thereby enhancing the quality and reliability of the obtained data. The refined nanoparticle suspension ensures more accurate and reproducible UV-Vis measurements by reducing baseline noise and minimizing scattering effects. Herein, the nanoparticle suspension was first centrifuged at 6000 RPM for 12 min, and the supernatant containing unwanted by-products was carefully removed by pipette. The resulting pellet was then redispersed in deionized water and subjected to four repeated centrifugation cycles to completely eliminate soluble impurities. In a subsequent step, the samples were washed with ethanol to facilitate rapid drying and minimize agglomeration. Finally, the purified samples were dried under ambient conditions to ensure the preservation of their structural integrity and subsequently analyzed using SEM (JEOL JSM-6710F FEG-SEM). UV-visible spectra of the collected samples were obtained using an INESA SHANGFEN L6S UV-visible spectrophotometer over the wavelength range of 400–700 nm.

## Numerical simulation

This part outlines the numerical simulation approach taken to determine the performance characteristics of the proposed hybrid passive micromixer. The primary criterion for selecting a design is its mixing performance, which can be evaluated through mathematical modeling of fluid dynamics and mass transport of a diluted species, irrespective of the nature of the diluted species.

In the context of fluid mechanics, it is well-established that microfluidic channels operate under laminar flow conditions [41], characterized by a low Reynolds number (Re) as a consequence of their small size and reduced flow rates [42]. In laminar microflows, fluid streamlines remain predominantly parallel, mixing primarily governed by molecular diffusion driven by concentration gradients.

The mixing of the two fluids within the microchannel was modeled under steady-state conditions, following a methodology consistent with previous studies [43,44]. Since the primary objective is to evaluate mixing performance across different micromixer designs, steady-state analysis provides an effective framework by capturing the established flow and concentration fields.

The evolution of mixing, in the absence of chemical reactions and under steady-state conditions, is often described by the following advection-diffusion equation:

$$u \cdot \nabla C = D\nabla^2 C \qquad (1)$$

where $D$ is the diffusion coefficient, $C$ is the concentration of the diluted species. The influence of the fluid's flow on the substance's transport is represented by the convective term on the left, showing how the velocity field causes the concentration to be advected. Conversely, the right-hand side's diffusion term captures the substance's tendency to spread from high-concentration zones to low-concentration zones.

The numerical simulations were performed using the commercial finite element software COMSOL Multiphysics. The Laminar Flow and Transport of Diluted Species interfaces were employed to model incompressible laminar flow and advection-diffusion driven species transport under low Reynolds number conditions. These modules are well-suited for modeling low-Reynolds-number flows and mass transport in microfluidic devices. For the numerical simulations, the hybrid micromixer design was modeled using the same geometry and dimensions as those employed in the experimental setup. Specifically, the design consisted of a Y-junction microchannel incorporating an embedded helical structure, precisely replicating the geometric parameters and dimensions outlined in the fabrication section.

Selecting an appropriate mesh size is critical because it governs both numerical accuracy and computational cost. Accordingly, a mesh-sensitivity assessment was performed for the hybrid micromixer prior to the production simulations. Mesh dependence was quantified at $Re = 1$ using the mixing index evaluated at a fixed downstream location $x = 6$ mm. Based on this analysis, a mesh consisting of approximately seven million elements was found to yield mesh-independent results, with changes of less than 2% upon further refinement. This mesh was therefore adopted for all reported simulations. Additional details of the mesh-sensitivity study, including quantitative comparisons and mesh illustrations, are provided in Fig S1 in the S1 File. The domain was discretized with tetrahedral elements, with quadratic discretization for enhanced precision, and a minimum element size of 0.5 μm was imposed. As illustrated in Fig 3, particular attention was paid to regions where the helix structure is close to the main channel walls. The mesh was refined in these areas to

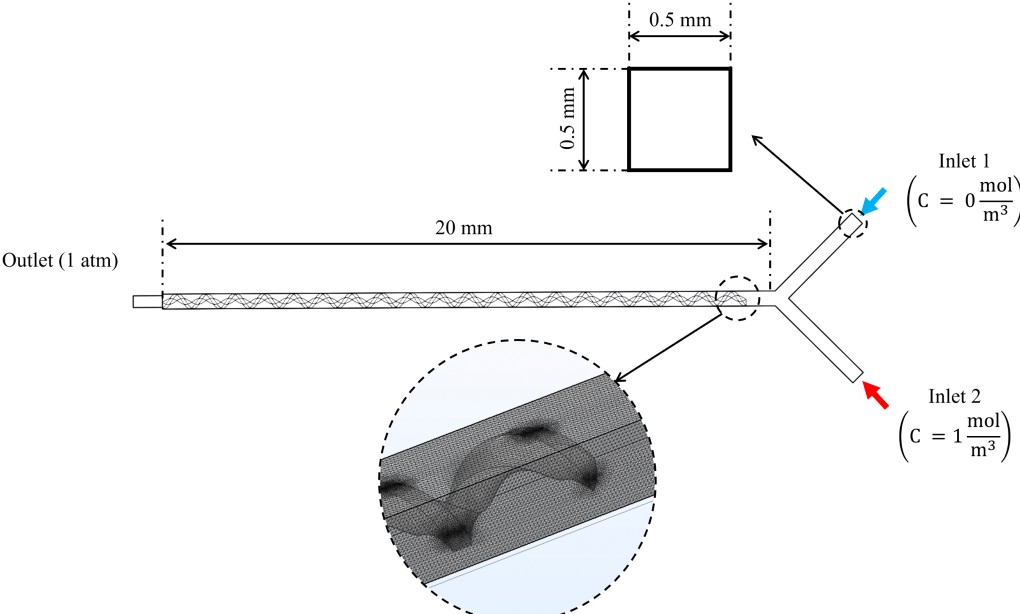

**Fig 3. The schematic representation of the hybrid device illustrates its geometry, dimensions, and computational mesh, which were used in the simulations conducted in this study.**

accurately capture the potentially high velocity and concentration gradients within the narrow gaps, ensuring the resolution of complex flow patterns and the overall accuracy of the simulation.

This numerical study utilized water as the fluid medium, consistent with the experimental setup. The water was modeled with a viscosity of 0.001 Pa·s and a density of 1000 kg/m³. As shown in Fig 3, identical flow rates were imposed at both inlets, with tested values of 30, 111, and 278 µL/min, matching those used in the experimental study. A no-slip boundary condition was applied at the channel walls to simulate fluid-solid surface interaction. The outlet boundary was set to atmospheric pressure. The Codina method, a stabilization technique used in finite element simulations to prevent numerical instabilities in convection-dominated problems by minimizing unphysical oscillations and ensuring stable pressure and velocity calculations in incompressible flow, was employed to account for crosswind diffusion, ensuring a realistic representation of the mixing process [45]. Finally, for the diluted species boundary conditions, one inlet was supplied with pure water (0 mol/m³ species concentration), while the other received water with a 1 mol/m³ concentration of the same species (see Fig 3). This concentration difference drove the mixing process within the device.

The numerical model employed in this study was validated against previously published work [43], ensuring its accuracy and reliability. Detailed comparison, including specific parameters, validation data, and relevant references, can be found in section 2 in the S1 File.

## Results and discussion

This section presents a comprehensive analysis of the results obtained from numerical simulations and experimental validations of the proposed hybrid micromixer. The discussion is organized into two primary sections. First, a detailed characterization of the mixing performance is presented; in this section, mixing efficiency is evaluated as a function of the Reynolds number, and comparisons are made with the performance of conventional micromixer designs. Second, the application of the micromixer for the controlled synthesis of gold nanoparticles is investigated; this section focuses on how flow rate affects nanoparticle size and distribution, as determined by UV-Vis spectroscopy and SEM imaging.

### Mixing characterization

The hybrid micromixer, combining a traditional Y-junction with a 3D-printed helical structure, has been subjected to a series of tests across three different Reynolds numbers (3.7, 18.2, and 37.04) to evaluate the efficacy of the hybrid micromixer in enhancing mixing performance. To quantify mixing efficiency in the numerical simulations, several cross-sectional planes normal to the main flow direction were defined at fixed axial locations along the mixing channel. The first cut plane was positioned at the inlet ($x = 0$), where the two fluid species initially come into contact, followed by additional planes at $x = 3, 6, 9, 12, 15, 18$, and 20 mm downstream of the inlet. At each cut plane, the spatial distribution of the solute species concentration across the channel cross-section was extracted, and the following formula was utilized:

$$\eta = 1 - \frac{\sigma}{\sigma_{max}}$$

(2)

In this expression, $\eta$ denotes the mixing performance, $\sigma$ refers to the standard deviation of the solute species concentration in the cut plane at a specific measurement point and $\sigma_{max}$ indicates the standard deviation of the concentration at the inlet ($x = 0$), corresponding to the fully unmixed initial condition. This approach enables a consistent evaluation of the progressive mixing performance along the axial direction of the micromixer.

This section presents the experimental results obtained from the physical prototype of the hybrid micromixer. The micromixer performance was examined under real-life conditions, and the outcomes were compared with the numerical predictions.

The numerical model is employed to compare the mixing performance of the hybrid microchannel with three conventional designs reported in the literature: the T-junction, Tesla-like, and caterpillar mixers. These designs were selected for

benchmarking because they represent widely used passive micromixers with distinct mixing mechanisms, making them suitable for assessing the advantages of the hybrid structure. To ensure a fair comparison, the geometries of these conventional mixers were replicated from a previous study [43] and simulated under identical conditions using the procedure outlined in Section 3. As shown in Fig 4, the hybrid design achieves a significant mixing efficiency of approximately 66% within the first 5 mm of travel, demonstrating substantial improvements over the T-junction (+56%), Tesla-like (+53%) and the caterpillar mixer (+47%). The design advantage of the hybrid micromixer is particularly effective at low to moderate Reynolds numbers, where the helical structure optimizes fluid flow and reduces diffusion lengths, leading to significantly improved mixing outcomes compared to traditional designs. After the initial phase, the mixing performance of the hybrid design continues to increase more modestly. Yet, the performance gap between the hybrid design and conventional mixers remains substantial throughout the entire 20 mm channel length. After 20 mm of travel, the hybrid micromixer achieves an efficiency of > 80%, which is considerably higher than those reported for the T-junction (+ 64%), Tesla-like (+ 47%) and caterpillar mixer (+ 35%).

The mixing performance of the proposed hybrid micromixer was benchmarked against representative passive micromixers reported in the literature. At a channel length of 5 mm and Re = 0.01, reported mixing indices for conventional passive designs range from 0.27 to 0.90, whereas the present hybrid micromixer achieves a mixing index of 0.99. At Re = 1, literature values typically fall between 0.13 and 0.54, while the proposed design maintains a higher mixing index of 0.73, demonstrating superior mixing performance under both diffusion-dominated and inertia-influenced conditions. A detailed comparison is provided in Table S1 in the S1 File.

Fig 5 (a, b) shows images of the hybrid microchannel captured during experiments performed at flow rates of 30 μL/min (a) and 111 μL/min (b), corresponding to *Re = 1* and 3.7, respectively. The helical structure's influence on mixing

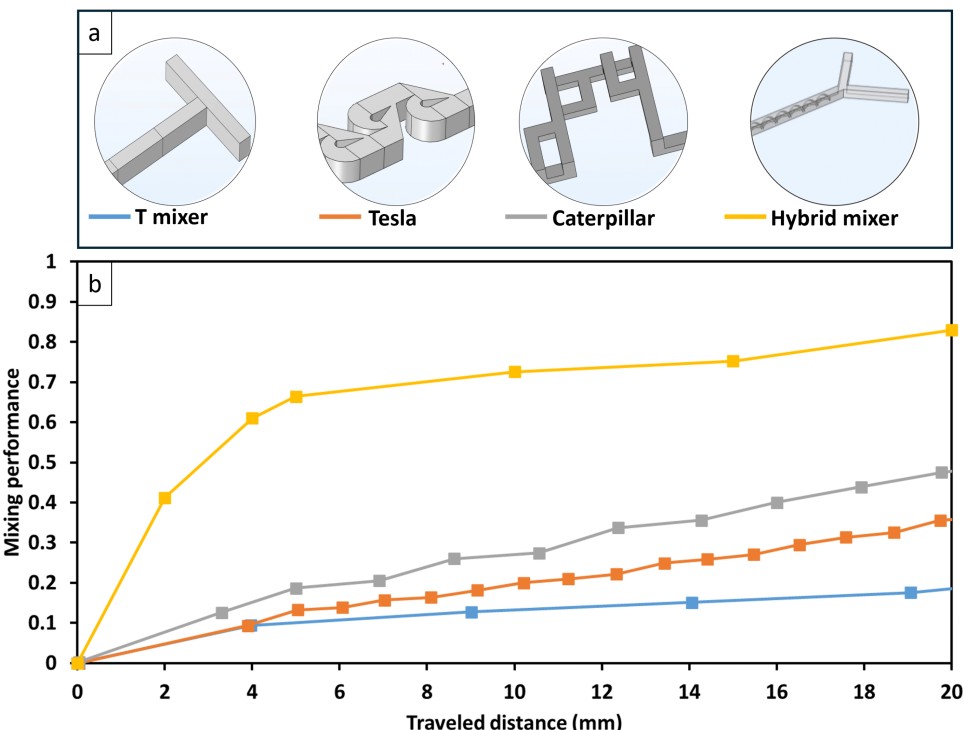

**Fig 4. Design and mixing performance of passive micromixers.** (a) Schematic illustration of the design of different passive micromixers. (b) Predicted mixing performance of the hybrid micromixer compared to conventional micromixer designs at Re = 3.7.

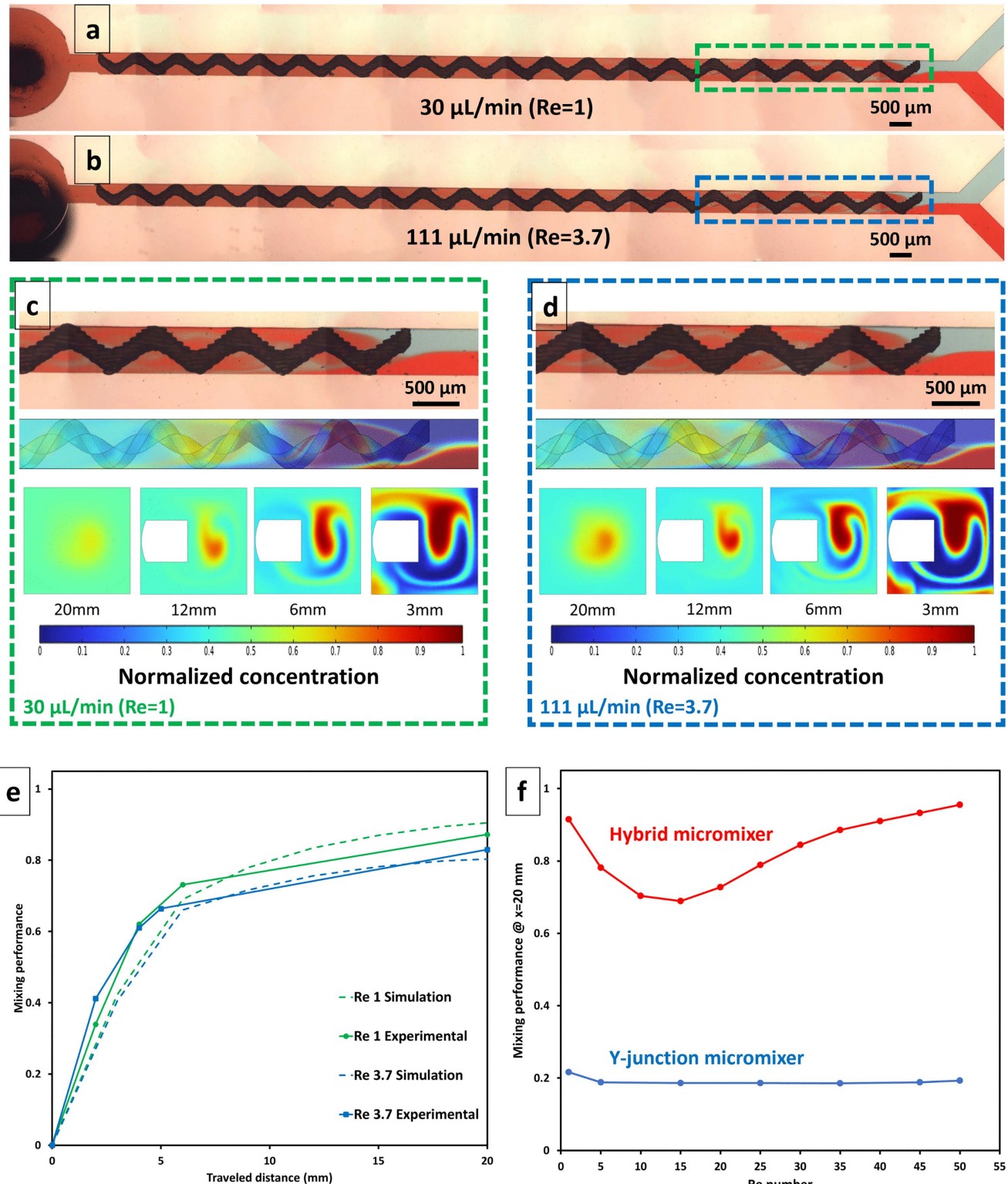

**Fig 5. Experimental and numerical evaluation of hybrid micromixer performance.** Images taken during experiments conducted at flow rates of 30 µL/min (a) and 111 µL/min (b) with magnified views (c, d) of the mixing region contrasted against numerical predictions of mixing efficiency, and spatial snapshots of the concentration contours at different cut planes. **(e)** Quantitative comparison of mixing performance between experimental and simulated results. **(f)** Predicted results of mixing performance of the hybrid micromixer compared to a conventional Y-junction micromixer (without the 3D element).

is visually apparent, as it induces a swirling motion within the flow, giving rise to eddies that play a significant role in enhancing the mixing process. The presence of these eddies indicates the dynamic interplay between the fluid streams, further facilitated by the helical attachment, which disrupts the laminar flow and promotes thorough mixing. Moreover, Fig 5 (c, d) present magnified views of the mixing region within the hybrid micromixer at flow rates of 30 µL/min (Re = 1) and 111 µL/min (Re = 3.7), respectively. The experimental images in the upper panels illustrate the flow evolution of the two-colored inlet streams, while the corresponding numerical simulations shown below depict the spatial distribution of the normalized concentration field. In the numerical model, one inlet stream is assigned a concentration of 1 mol/m³ and the other 0 mol/m³; therefore, the resulting mixed concentration lies between 0 and 1 mol/m³, where a normalized concentration value of 0.5 corresponds to fully mixed conditions (100% mixing efficiency). The appearance of intermediate concentration values (yellow regions) at early axial locations reflects rapid onset of mixing rather than partial mixing. In addition, snapshots of the concentration contours are provided at several downstream channel lengths for both flow conditions, enabling a direct comparison of the mixing evolution along the micromixer. Consistent with Fig 4, the concentration contours demonstrate progressive homogenization along the channel length, with complete mixing achieved downstream as chaotic advection induced by the helical structure generates strong rotational flow, eddies, and enhanced transverse transport.

Fig 5(e) presents the mixing performance of the hybrid microchannel across two flow rates, corresponding to Reynolds numbers of 1 and 3.7 for both experiments and simulations. The results demonstrate strong agreement between the experimental findings and the simulation outcomes, indicating a rapid increase in mixing within the initial 6 mm, followed by a more gradual progression. The maximum deviation between the simulation and experiment was 6% at the outlet, which occurred at Re = 1. The slight drop in mixing performance observed at Re = 3.7 compared to Re = 1 can be attributed to the reduced residence time of the fluid streams, which consequently limits mass transfer by diffusion. The consistency between experimental results and simulation data confirms the accuracy of the simulations in capturing these crucial aspects of fluid dynamics, which is essential for optimizing microfluidic designs for enhanced mixing.

Fig 5(f) compares the predicted mixing performance of the hybrid passive micromixer (red) to that achieved by the conventional Y-junction microchannel without the helix included (blue) at a downstream position of 20 mm across Re = 1–50. The straight Y-junction is included here as a baseline reference geometry to isolate and quantify the mixing enhancement introduced by the embedded 3D-printed helical structure, rather than as a benchmark against state-of-the-art passive micromixers. The hybrid design demonstrates superior performance, maintaining efficiency above 70% over the tested range of flow rates. In comparison, the Y-junction microchannel experiences a slight decline in mixing efficiency as the Reynolds number increases from 1 to 5, after which the efficiency remains nearly constant over the remaining range (Re = 5–50). This is due to reduced fluid residence time, which limits molecular diffusion at higher flow velocities. Comparative benchmarking against more advanced passive micromixer designs, including Tesla-like and caterpillar mixers, is presented separately in Fig 4 and summarized in Table S1 in the S1 File. Experimental observations of fluid streams in the Y-junction micromixer at different Re numbers are presented in S4 Fig in the S1 File. Notably, at Re = 1, the hybrid micromixer design achieves a superior mixing efficiency of 92%. As the Reynolds number increases further to intermediate values (Re is 5–15), a gradual decline in mixing efficiency is observed. However, with a further increase in the Reynolds number beyond Re = 15, the mixing efficiency improves steadily and reaches values above 95% at higher Reynolds numbers (approximately 40–50). The observed trend in mixing performance can be attributed to the balance between diffusion-dominated and convection-dominated mixing within the microchannel. At low Reynolds numbers, molecular diffusion serves as the primary mixing mechanism, and the low flow velocity allows sufficient time for diffusion to enhance mixing efficiency. However, as Re increases from 5 to 15, the velocity rises, reducing the residence time of fluid elements within the channel. Since the flow remains laminar, convective mixing remains relatively weak, and molecular diffusion alone becomes insufficient to maintain high mixing efficiency, leading to a decline in performance. With a further increase in Re to 50, inertial effects become more pronounced, promoting the development of secondary flow structures or mild

disturbances that enhance mixing. This improvement is likely due to the formation of Dean vortices or chaotic advection, which facilitate greater fluid interaction and improve mixing efficiency despite the higher flow rate.

## Gold nanoparticle synthesis

The hybrid micromixer was used to synthesize gold nanoparticles, which requires a thorough mixing between the gold precursor, $HAuCl_4$ (chloroauric acid), and the reducing agent, L-ascorbic acid (LASC). The helical geometry ensures efficient interaction between the reactants, facilitating the reduction of $Au^{3+}$ ions to $Au^0$, leading to the formation of gold nanoparticles. The precursor and the reducing agent are introduced into the device through separate inlets at equal flow rates, with concentrations set at 1 mM for each. Flow rates of 200, 500, 1000, 2000, and 5000 µL/min were tested to optimize the synthesis process.

The UV-Vis spectra of gold nanoparticles synthesized at varying flow rates demonstrate a clear correlation between flow rate, surface plasmon resonance (SPR) peak position, and peak broadness, as shown in Fig 6. At the highest flow rate (5000 µL/min, blue line), the SPR peak is sharp and centered at 528.5 nm, indicative of smaller nanoparticles with a narrow size distribution. As the flow rate decreases, the peaks progressively move toward longer wavelengths, which are closer to the red region of the visible spectrum to 536.5 nm (2000 µL/min, yellow), 548 nm (1000 µL/min, green), 547.5 nm (500 µL/min, orange), and 553 nm (200 µL/min, black), consistent with larger nanoparticle sizes or aggregation at slower flow rates. Concurrently, the peaks broaden significantly at lower flow rates (e.g., 200 µL/min), suggesting increased polydispersity or aggregation due to extended growth times. Notably, the near-identical peak positions for 500 µL/min (547.5 nm) and 1000 µL/min (548 nm) hint at potential kinetic limitations or experimental variability at intermediate flow rates, warranting further study. These findings suggest that the flow rate is a crucial parameter for controlling nanoparticle size and uniformity, with higher flow rates being optimal for applications requiring small, monodisperse particles. Lower flow rates might be preferred when larger particles are desired, albeit with less precise size control. This understanding is particularly valuable for optimizing synthesis conditions to achieve gold nanoparticles with specific size characteristics for targeted applications.

SEM images of gold nanoparticles synthesized at a flow rate of 500 µL/min reveal particles with significant size variations, as shown in Fig 7 (a-b). The particles exhibit irregular, quasi-spherical morphologies with sizes ranging from 30 to 127 nm. Such a broad size distribution correlates well with the previous UV-Vis data in Fig 6, where the 500 µL/min sample showed a broader absorption peak, a direct consequence of the polydispersity observed in the SEM images. The formation of such varied particle sizes at this flow rate (500 µL/min) can be attributed to the relatively slower reaction kinetics, allowing different growth stages to occur simultaneously. Some particles have more time to grow while new nucleation sites are still forming, resulting in the observed heterogeneous size distribution.

The SEM images of gold nanoparticles synthesized at 2000 µL/min are presented in Fig 7 (d, e), showing a more monodisperse size distribution than the sample synthesized at 500 µL/min. The particles are predominantly spherical, with smooth surfaces and diameters ranging from approximately 20–50 nm. This narrow size distribution aligns with the sharper, more intense UV-Vis absorption peak observed for the 2000 µL/min sample. The higher flow rate enables faster mixing and more homogeneous reaction conditions, promoting uniform nucleation and growth of the nanoparticles. The smaller average size at 2000 µL/min can be attributed to the shorter residence time, which limits particle growth.

Fig 7 (g, h) presents the SEM images of gold nanoparticles synthesized at a high flow rate of 5000 µL/min, showcasing well-defined quasi-spherical nanoparticles with a remarkably narrow size distribution averaging 14–25 nm. This exceptional mono-dispersity elegantly supports our earlier UV-Vis spectroscopic data, where the sharp absorption peaks indicated uniform particle populations. The synthesis kinetics at this elevated flow rate proved crucial, as the rapid flow facilitated simultaneous nucleation events while minimizing variations in growth periods. This uniformity is illustrated across different scales; the lower magnification image, Fig 7(g), provides a comprehensive view of the particle population's consistency, while the higher magnification image, Fig 7(h), reveals intricate details of individual particle morphology,

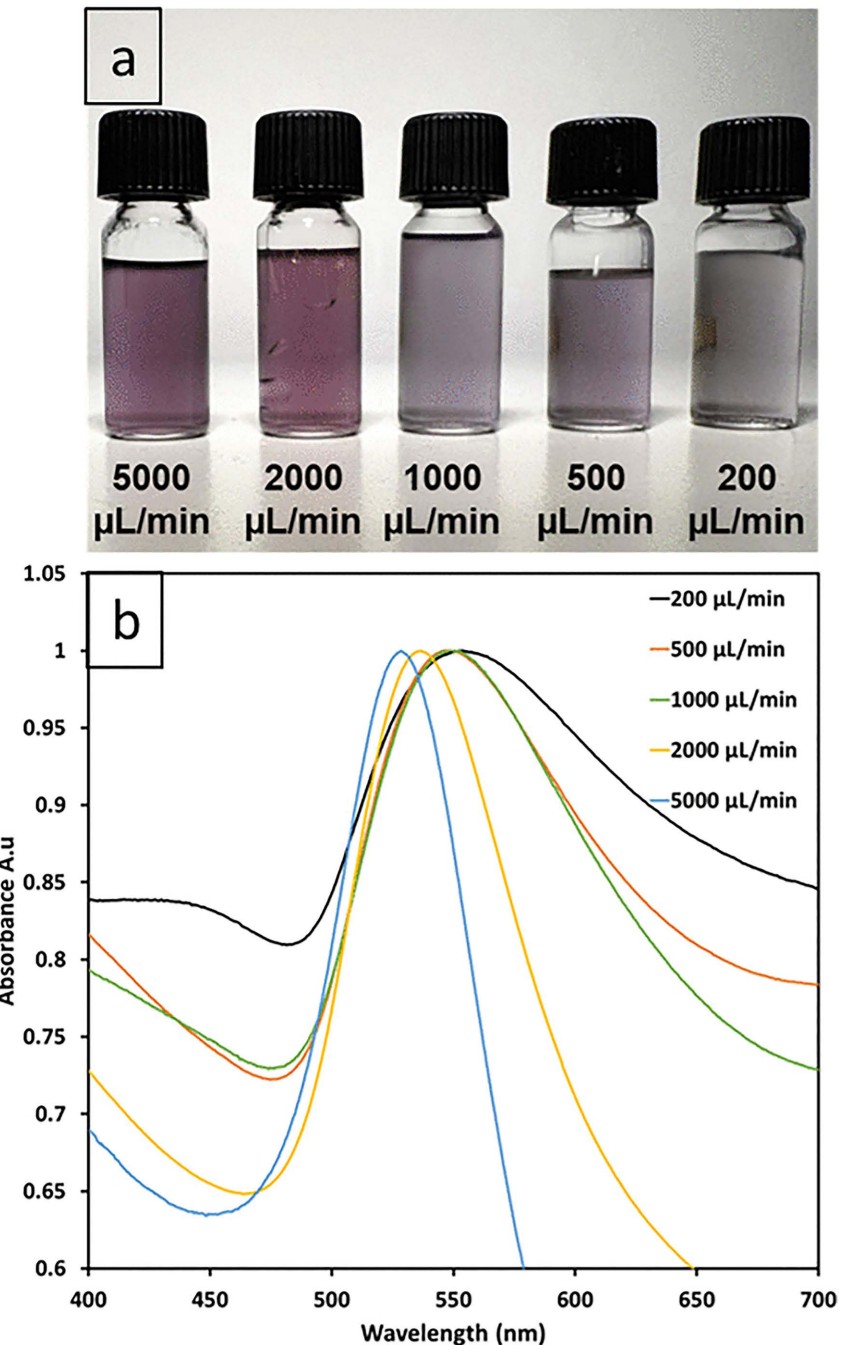

**Fig 6. UV-Vis analysis of collected samples at different flow rates.** (a) Collected samples at different total flow rates (200, 500, 1000, 2000, 5000 µL/min), (b) UV-Vis spectra of five different total flow rates.

both confirming the exceptional size control achieved. The microscopic evidence firmly establishes that elevated flow rates in microfluidic synthesis create ideal conditions for producing monodisperse gold nanoparticles through precise control over nucleation and growth mechanisms, marking a significant advancement in the synthesis of uniform nanomaterials for precise applications.

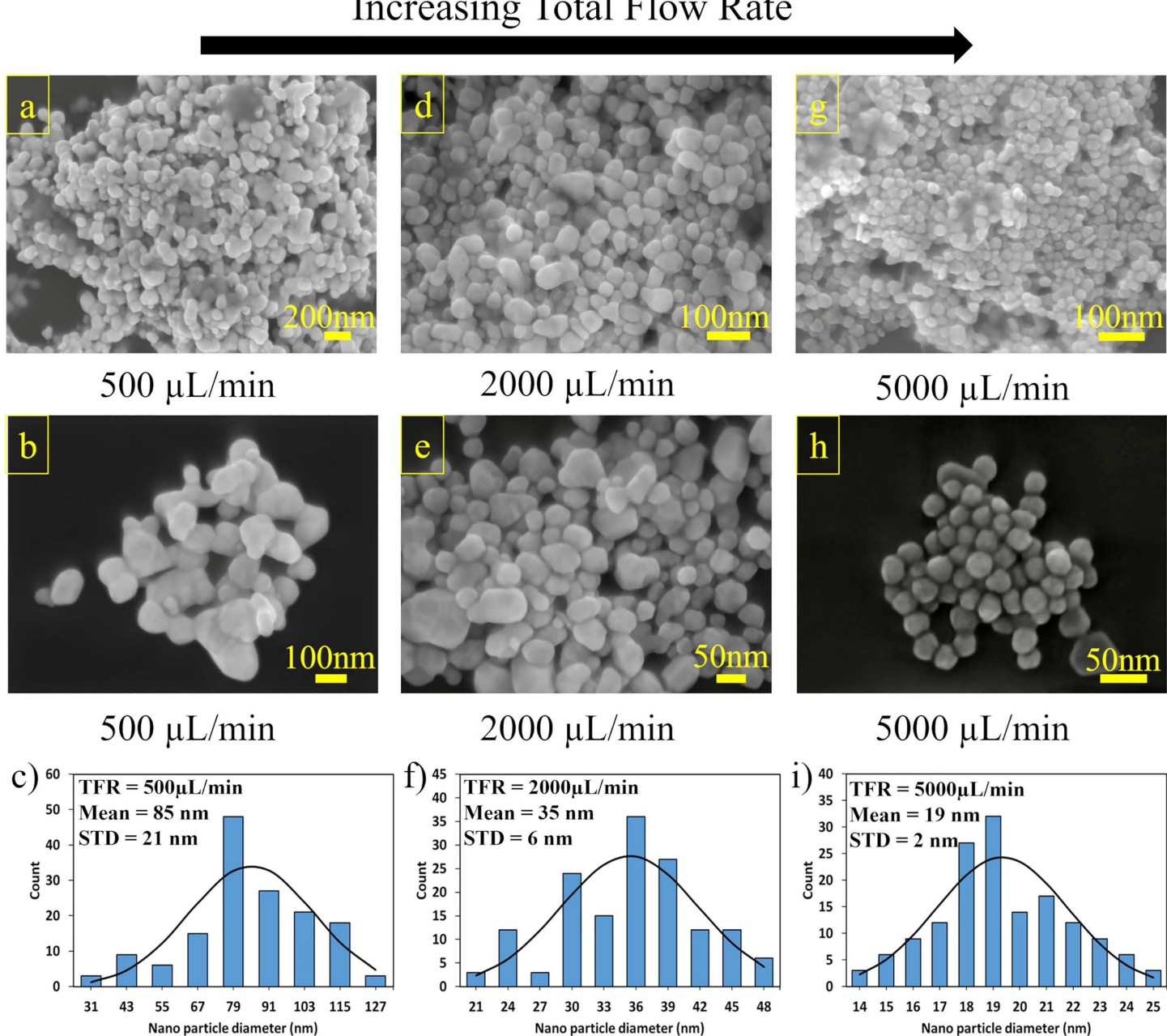

**Fig 7. Effect of Total Flow Rate on the Morphology and Size Distribution of Gold Nanoparticles.** SEM images of gold nanoparticles synthesized at total flow rates of 500 μL/min (a, b), 2000 μL/min (d, e), and 5000 μL/min (g, h) at different magnifications. Corresponding particle size distributions obtained from SEM analysis are shown for total flow rates of 500 μL/min (c), 2000 μL/min (f), and 5000 μL/min (i), with mean particle diameters and standard deviations indicated.

Particle diameters were measured in ImageJ after pixel-to-nanometer calibration; for each condition, n = 150 nanoparticles were sampled from multiple, non-overlapping fields of view, and the corresponding size histograms are provided in Fig 7. The distributions were unimodal and shifted to smaller sizes with increasing flow rate, yielding 85 ± 21 nm at 500 μL/min, 35 ± 6 nm at 2000 μL·min⁻¹, and 19 ± 2 nm at 5000 μL·min⁻¹ (mean ± STD). A progressive narrowing of the distributions

at higher flow verifies the SEM observations and is consistent with the UV–Vis spectra, which exhibit a blue shift and peak narrowing of the surface-plasmon resonance with increasing flow rate. Operationally, no fouling or clogging was observed. Moreover, after each run, a brief gold-etchant rinse was applied to remove residual nanoparticle sediment from the microchannel and 3D-printed structure, maintaining stable performance throughout.

## Conclusion

This study developed a hybrid passive micromixer by integrating traditional soft lithography microfabrication techniques with cost-effective 3D printing to enhance mixing efficiency for nanoparticle synthesis. The microchannel was fabricated from PDMS using soft lithography, and a 3D-printed helical structure, produced with an affordable DLP printer, was embedded within the channel. Additionally, numerical simulations were conducted to optimize the micromixer design and operational parameters, with the helical structure demonstrating superior mixing capabilities under laminar flow conditions. Experimental validation was performed using dyed fluids, and mixing efficiency was quantitatively assessed through MATLAB-based image analysis. The hybrid micromixer exhibited enhanced mixing performance, with results aligning closely with simulation predictions. Notably, the hybrid micromixer achieved a mixing efficiency of 92% at Re = 1 and 83% at Re = 3.7. This significantly surpasses the performance of conventional passive micromixers at Re = 3.7, where a T-mixer achieves only 19% mixing efficiency (a 337% improvement by our design), a Tesla-like mixer reaches 39% (a 113% improvement), and a caterpillar mixer achieves 55% (a 51% improvement).

Gold nanoparticles were synthesized within the micromixer using L-ascorbic acid as the reducing agent, and the products were characterized through UV-Vis spectroscopy and SEM. The synthesized gold nanoparticles ranged in size from 14 to 25 nm at a total flow rate of 5000 μL/min, demonstrating precise control over the nanoparticle size distribution. This hybrid micromixer approach provides a scalable, efficient, and cost-effective platform for nanoparticle synthesis, with potential applications in nanotechnology and materials science.

Operational robustness was also evaluated. Across all runs, including the highest tested flow rate of 5000 μL/min, no fouling, clogging, leakage, or delamination was observed. The permanent bonding and smooth, optically clear channel walls helped suppress particle deposition, and a brief gold-etchant rinse after each run effectively removed residual nanoparticles from the microchannel and the helical structure, restoring baseline conditions.

Despite the promising results, this study has several limitations. The current work focused exclusively on gold nanoparticle synthesis using L-ascorbic acid, leaving the micromixer's performance with other materials and reactions unexamined. Furthermore, the size of the 3D-printed structure is constrained by the minimum resolution of the DLP printer, which may limit the design's scalability and versatility.

## Supporting information

**S1 File. Supplementary information including mesh independence study, numerical model validation, comparison of mixing performance of passive micromixers reported in the literature, and Y-shaped micromixer performance.**
(DOCX)

**S2 File. Data.**
(XLSX)

## Author contributions

**Conceptualization:** Anas Alazzam.

**Data curation:** Yasser Aldaghestani, Anas Alazzam.

**Formal analysis:** Yasser Aldaghestani, Andreas Schiffer.

**Funding acquisition:** Anas Alazzam.

**Methodology:** Andreas Schiffer.

**Project administration:** Anas Alazzam.

**Resources:** Anas Alazzam.

**Software:** Yasser Aldaghestani.

**Supervision:** Andreas Schiffer, Anas Alazzam.

**Validation:** Yasser Aldaghestani.

**Visualization:** Yasser Aldaghestani.

**Writing – original draft:** Yasser Aldaghestani.

**Writing – review & editing:** Andreas Schiffer, Anas Alazzam.

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
