## [Decision Letter · Decision Letter 0]

1 Dec 2025

Dear Dr. Alazzam,

Thank you for submitting your manuscript to PLOS ONE. After careful consideration, we feel that it has merit but does not fully meet PLOS ONE’s publication criteria as it currently stands. Therefore, we invite you to submit a revised version of the manuscript that addresses the points raised during the review process.

Please submit your revised manuscript by Jan 15 2026 11:59PM. Please respond to all comments from all reviewers. If you will need more time than this to complete your revisions, please reply to this message or contact the journal office at plosone@plos.org . A rebuttal letter that responds to each point raised by the academic editor and reviewer(s). You should upload this letter as a separate file labeled 'Response to Reviewers'.A marked-up copy of your manuscript that highlights changes made to the original version. You should upload this as a separate file labeled 'Revised Manuscript with Track Changes'.An unmarked version of your revised paper without tracked changes. You should upload this as a separate file labeled 'Manuscript'.

We look forward to receiving your revised manuscript.

Kind regards,

Bonnie Gray

Academic Editor

PLOS ONE

Journal Requirements:

[This work was supported by the System on Chip Lab, Department of Mechanical and Nuclear Engineering, Khalifa University of Science & Technology.]

[The author(s) received no specific funding for this work.]

4. Your striking image file will represent your article upon publication on the PLOS ONE homepage. The image must be derived from a figure or supporting information file from your manuscript. Ideally, striking images should be high resolution, eye-catching, single panel images that do no contain additional text, scale bars, or arrows.

Please also keep in mind that PLOS's Creative Commons Attribution License applies to striking images. As such, please do not submit any figures or photos that have been previously copyrighted unless you have express written permission from the copyright holder to publish under the CCAL license. You can read more about PLOS’s Creative Commons License on our homepage: http://journals.plos.org/plosone/s/licenses-and-copyright

5. Please include captions for your Supporting Information files at the end of your manuscript, and update any in-text citations to match accordingly. Please see our Supporting Information guidelines for more information: http://journals.plos.org/plosone/s/supporting-information .

Reviewers' comments:

Reviewer's Responses to Questions

**Comments to the Author**

1. Is the manuscript technically sound, and do the data support the conclusions?

Reviewer #1: Yes

Reviewer #2: Yes

2. Has the statistical analysis been performed appropriately and rigorously?

Reviewer #1: N/A

Reviewer #2: Yes

3. Have the authors made all data underlying the findings in their manuscript fully available?

Reviewer #1: Yes

Reviewer #2: Yes

4. Is the manuscript presented in an intelligible fashion and written in standard English?

Reviewer #1: Yes

Reviewer #2: Yes

Reviewer #1: Review: PONE-D-25-52946 - Hybrid Passive Micromixer Using Combined Traditional Microfabrication and 3D Printing for Gold Nanoparticle Synthesis

The authors did a nice research on different aspects of the synthesis of gold nanoparticles (AuNPs) through a hybrid microfluidic device. They used a combined fabrication method for a microfluidic chip, soft-lithography and DLP for the Y-junction mixer including helical structure to enhance mixing. To support the design optimization they ran numerical simulations and experimental validation, and reached a mixing efficiency of 92% at a Reynolds number of 1 by using passive mixers to achieve mixing at low Reynolds numbers.

The manuscript is not accepted at this stage and needs to be revised based on the following comments. It is suggested that the authors keep the nature of the research paper different from a review paper and need to address point-by-point answers for the following comments.

Introduction

The introduction needs to be rewritten. It is too long and not categorized well for different topics of microfluidics, passive, active mixers, current limitations, advantages of microfluidics for NPs applications, fabrication methods and the role of 3D printing to improve mixing, geometry of micromixers, etc…. You don’t have to talk about all in detail like a review paper. Combine ones in shorter sentences and highlight the contribution of the paper. Avoid redundancy and back and forth discussion, at the very end of introduction you talked about passive micromixers again! The content of such interdisciplinary topic is too much for a research paper. From fabrication, to fluid mechanics, to synthesis, types of mixers, etc.!

The second paragraph of the introduction jumps from active/passive micromixer to the synthesis of nanoparticles. A transition is needed. Also, the following paragraph starting with However, needs to be followed in the same paragraph.

DLP-Softlithography is a better term. Parallelism for technology instead of DLP-PDMS (one technology the other material)

Design and fabrication

hybrid passive micromixer: Hybrid in terms of what?

Adding two dashed blocks in Figure-1 for PDMS fabrication AND DLP printing would makes the fab procedure more clear in a glance.

Section 2.3

In the 3rd line you said PDMS microchannels. Please add the material for the DLP printed helical structure too. What is the backbone of the resin? This would clarify the next discussions about PDMS-PDMS bonding.

How did the authors manage alignment for embedding the helix structure in the microchannels? What setup adjustment was taken using Dilase? I see you used A 50 μm stop feature. Label that in Fig2-b and elaborate how that helps prevent axial migration? Instead of insertion into microchannels, couldn’t you directly 3D print the helical structure on the PDMS microchannel??

It’s better to add the target resolution when DLP and 2PP are compared, in the paragraph before section 2.4

Table 1 could be moved to the supplementary and the best efficiencies be cited in one sentence in the main text. This is not a review paper.

Figure 5 shows the max of mixing performance happens at 20 mm, whereas the concentration contours in Fig6 d shows the samples are mixed to 30% of the initial concentration after 4mm. Figure 6C show two yellow (35%) at two locations. Please clarify and also provide spatial snap shots of the concentration at different locations along the channel.

Figure 6f: What is the justification for the minimum of mixing performance at Re=20?? If the authors attribute it to the diffusion-dominated vs convection-dominated flow at this point, have you looked at the concentration contours at Re 20?

What is the coordinate/location of the cut plane? It’s important to show that/mention that for the performance graphs. Additional x(or y) information along the mixing channel is required for “a cross-sectional plane was established after each mixing element…” and “solute species' concentration in the cut plane at a specific measurement point”. What is the coordinates of that point???

Figure 7 needs to be revised for the legend, units of flowrates on the graph although mentioned in the caption.

Figure 8: One of the SEM scale bars in “d” or “e” must be wrong for the same flow rate of 2000ul/min.

Is the optimized synthesis of AuNPs, ranging from 12‒30 nm, for a total flow rate of 5000 μl/min?

Can the authors introduce a parameter based on the size and flow rate for a more generic NPs synthesis?

Any statistical analysis for the printed chips?

Editorial

Proof read for typos. (Typo : neMESYS syringe pump , missing period after “and reduced transparency” )

Reviewer #2: This paper details a method for producing laminar-flow in-plane and out-of-plane mixers using a mixture of conventional microlithography and 3D Printing. It makes a point for helical channels for improved mixing which required a stereolithography processing step on top of the microlithographically produced channel. The paper is relatively straight-forward in manner and the techniques demonstrated have both been demonstrated before, albeit not with specifically this focus. The authors chose a bit unconventional application scope for this channel setup, i.e., the reduction synthesis of gold nanoparticles which requires mixing two solutions (a metal salt and the reductant).

A couple of suggestions should be addressed before considering this manuscript for publication.

Remove the section on the NS equation - this can be referenced as the equation is not required here. We know which equation the solver needs to solve. As this is a case of diffusive mixing, Talor-Aris dispersion should be considered here more so than the fundamental simplified NS equation. However, neither of this is relevant for the manuscript at hand.

Mixing performances of the designs is compared numerically only and references to experimental validations are given. However, Figure 6f should compare the performance to a Tesla mixer rather than a straight Y-channel which is not used for mixing so much anymore.

Lastly, please compare the particle synthesis against a macroscopic experiment in a stirred vial which is de facto standard of how these syntheses are conducted today. Here, the diameter variation can only be varied by stoichiometry and not by flow rates which is a strong reason for choosing the microfluidic approach.

Overall, this is an interesting study. Albeit not ground-breaking, it will surely find its place in the literature as it shows an interesting advantage of hybrid manufacturing in microfluidics.

The reviewer suggest publications after revision.

**Do you want your identity to be public for this peer review?** For information about this choice, including consent withdrawal, please see our Privacy Policy

Reviewer #1: No

Reviewer #2: **Yes:** Bastian E. Rapp

---

## [Author Response · Author response to Decision Letter 1]

15 Jan 2026

Reviewer#1, Comment # 1: The introduction needs to be rewritten. It is too long and not categorized well for different topics of microfluidics, passive, active mixers, current limitations, advantages of microfluidics for NPs applications, fabrication methods and the role of 3D printing to improve mixing, geometry of micromixers, etc…. You don’t have to talk about all in detail like a review paper. Combine ones in shorter sentences and highlight the contribution of the paper. Avoid redundancy and back and forth discussion, at the very end of introduction you talked about passive micromixers again! The content of such interdisciplinary topic is too much for a research paper. From fabrication to fluid mechanics, to synthesis, types of mixers, etc.!. The second paragraph of the introduction jumps from active/passive micromixer to the synthesis of nanoparticles. A transition is needed. Also, the following paragraph starting with However, needs to be followed in the same paragraph. DLP-Softlithography is a better term. Parallelism for technology instead of DLP-PDMS (one technology the other material).

Author response: We thank the reviewer for their insightful comments and helpful suggestions. The Introduction has been substantially revised, condensed, and reorganized to improve clarity, coherence, and focus, while avoiding review-style depth and redundancy. The revised Introduction now follows a clear logical flow: the first paragraph introduces the general advantages and applications of microfluidic systems; the second paragraph focuses on micromixing challenges under laminar flow and distinguishes between active and passive micromixers; the third paragraph narrows the discussion to passive micromixers for nanoparticle synthesis and outlines current limitations; the fourth and fifth paragraphs introduce additive manufacturing and helical/spiral geometries as effective passive mixing strategies; the sixth paragraph reviews recent 3D-printed micromixers and identifies scalability limitations of high-cost 2PP approaches; and the final paragraphs clearly highlight the contribution of this work by presenting a cost-effective hybrid DLP-soft lithography micromixer for gold nanoparticle synthesis. In addition, terminology has been corrected to “DLP-soft lithography,” transitions between topics have been improved, and redundant discussions have been removed. Please see the revised Introduction on pages 3-5.

Reviewer#1, Comment # 2: Hybrid in terms of what?

Author response: We thank the reviewer for pointing out the need for clarification. The term “hybrid” refers to the integration of two fabrication approaches, namely a PDMS microchannel fabricated by soft lithography and a DLP-printed helical insert that induces chaotic advection in addition to diffusive mixing. This definition has now been explicitly explained in the revised introduction section, please see page 5 lines 7-9.

Reviewer#1, Comment # 3: Adding two dashed blocks in Figure-1 for PDMS fabrication AND DLP printing would makes the fab procedure more clear in a glance.

Author response: We thank the reviewer for this constructive suggestion. Figure 1 has been updated to visually separate the PDMS soft-lithography process and the DLP-based fabrication of the helical structure using dashed outlines. Please see figure 1 in the revised manuscript.

Reviewer#1, Comment # 4: Section 2.3 In the 3rd line you said PDMS microchannels. Please add the material for the DLP printed helical structure too. What is the backbone of the resin? This would clarify the next discussions about PDMS-PDMS bonding.

Author response: We thank the reviewer for this helpful comment. Section 2.3 has been revised to explicitly describe the material of the DLP-printed helical structure. The manuscript now states that the microchannel consists of a PDMS channel and a DLP-printed helical structure fabricated from a commercial acrylate-based UV-curable photopolymer resin forming a crosslinked polymer network. Please check page 7 line 10-13.

Reviewer#1, Comment # 5: How did the authors manage alignment for embedding the helix structure in the microchannels? What setup adjustment was taken using Dilase?

Author response: We thank the reviewer for this comment. Alignment of the helical structure within the PDMS microchannel was achieved through a design based self-alignment approach rather than an external alignment setup. The channel cross-section and length were designed to closely match the outer diameter and length of the 3D-printed helix, enabling a self-centering interference fit during insertion. The structure was manually inserted using tweezers, and correct placement was verified under an optical microscope prior to sealing the device. Additional details have been included in the revised manuscript to clarify this procedure (page 8, line 5-13).

Reviewer#1, Comment # 6: I see you used A 50 μm stop feature. Label that in Fig2-b and elaborate how that helps prevent axial migration?

Author response: We thank the reviewer for this helpful suggestion. Figure 2b has been revised to explicitly label the 50 µm width reduction at the downstream end of the channel. In addition, Section 2.3 has been expanded to clarify the mechanical role of this constriction. Please check page 8 line 13 to page 9 line 1. ________________________________________

Reviewer#1, Comment # 7: Instead of insertion into microchannels, couldn’t you directly 3D print the helical structure on the PDMS microchannel??

Author response: We thank the reviewer for this insightful question. While direct DLP printing of the helical structure onto a PDMS substrate could be considered, it presents several practical challenges, including unreliable adhesion between the cured resin and PDMS, surface contamination that hinders oxygen plasma activation and bonding to the microchannel layer, and difficulty in removing support material without damaging the substrate. In addition, direct printing inside the microchannels is not feasible, as DLP printing requires a planar build surface and optical access, and uncured resin cannot be reliably removed from the microchannel. For these reasons, the helical structure was fabricated separately and subsequently integrated into the PDMS microchannel.

Reviewer#1, Comment # 8: It’s better to add the target resolution when DLP and 2PP are compared In the paragraph before section

Author response: We thank the reviewer for this helpful suggestion. The manuscript has been revised to explicitly include the typical resolution ranges of both DLP (25-50 µm) and two-photon polymerization (100-200 nm), and to clarify that the resolution provided by DLP is sufficient for the target geometry used in this work, whereas the higher resolution of 2PP is not required. Please see the revised text on page 10, line 7-12.

Reviewer#2, Comment # 9: Table 1 could be moved to the supplementary and the best efficiencies be cited in one sentence in the main text. This is not a review paper.

Author response: We thank the reviewer for this constructive suggestion. Following this recommendation, Table 1 has been moved to Supplementary Information (now Table S1). The main text has been revised to concisely summarize only the key benchmarking results by citing the best reported mixing efficiencies from the literature and directly comparing them with the performance of the proposed hybrid micromixer. Please see the revised text on page 19, line 5-11.

Reviewer#1, Comment # 10: Figure 5 shows the max of mixing performance happens at 20 mm, whereas the concentration contours in Fig6 d shows the samples are mixed to 30% of the initial concentration after 4mm. Figure 6C show two yellow (35%) at two locations. Please clarify and also provide spatial snap shots of the concentration at different locations along the channel.

Author response: We thank the reviewer for pointing out this ambiguity and apologize for the confusion caused by not clearly defining the normalized concentration scale in the original manuscript. In the numerical model, one inlet stream is assigned a concentration of 1 mol/m³ and the other 0 mol/m³; therefore, a normalized concentration of 0.5 corresponds to fully mixed conditions (100% mixing efficiency). This definition has now been explicitly added to the manuscript, and Figure 5 has been revised to include a normalized concentration color bar, and spatial snapshots of the concentration contours at different cut planes for both cases. Please see the revised text on page 20, line 7-21.

Reviewer#1, Comment # 11: Figure 6f: What is the justification for the minimum of mixing performance at Re=20?? If the authors attribute it to the diffusion-dominated vs convection-dominated flow at this point, have you looked at the concentration contours at Re 20?

Author response: We thank the reviewer for this thoughtful question. As discussed in the manuscript, the local minimum in mixing performance around Re=15 is attributed to the balance between diffusion-dominated and convection-influenced mixing. At this Reynolds number, the increase in flow velocity reduces residence time, while inertial effects and secondary flow structures induced by the helical geometry are not yet sufficiently developed to compensate, leading to a temporary reduction in mixing efficiency. Regarding the concentration contours, these were necessarily examined, as the mixing efficiency is directly computed from the spatial distribution of solute concentration extracted on multiple cross-sectional planes along the channel. The contours at Re =15 show less effective transverse homogenization compared to both lower Re (diffusion-dominated) and higher Re (inertia-assisted) cases, consistent with the observed trend in Figure 5f. To better resolve this behavior and confirm the trend, we have extended the analysis to a wider Reynolds number range (Re=0-50) and investigated more Re numbers, as shown in the updated Figure 5f. We believe this explanation clarifies the physical origin of the observed minimum. Please see the revised text on page 22 line 13-22.

Reviewer#1, Comment # 12: What is the coordinate/location of the cut plane? It’s important to show that/mention that for the performance graphs. Additional x(or y) information along the mixing channel is required for “a cross-sectional plane was established after each mixing element…” and “solute species' concentration in the cut plane at a specific measurement point”. What is the coordinates of that point???

Author response: We thank the reviewer for this comment. The manuscript has been revised to explicitly define the locations of the cross-sectional cut planes used to evaluate mixing efficiency. The axial positions of all measurement planes along the channel are now clearly stated, including the inlet reference plane. Please refer to the revised text on page 17, line 12 to page 18, line 3 for details.

Reviewer#1, Comment # 13: Figure 7 needs to be revised for the legend, units of flowrates on the graph although mentioned in the caption.

Author response: We thank the reviewer for this suggestion. Figure 6 has been revised to explicitly include the flow-rate units directly in the plot legend. Each curve is now labeled with its corresponding total flow rate including units (µL/min). Please see the revised figure on page 25.

Reviewer#1, Comment # 14: Figure 8: One of the SEM scale bars in “d” or “e” must be wrong for the same flow rate of 2000ul/min.

Author response: We thank the reviewer for carefully examining Figure 7. The scale bars in panels (d) and (e) were originally correct and reflected images acquired at different SEM magnifications for the same flow rate (2000 µL/min). Consequently, the scale bar in panel (e) appears larger relative to the field of view than in panel (d), which is consistent with the increased magnification. However, to improve clarity and maintain visual consistency with the SEM images presented for other flow rates, we have revised the scale bar in panel (e) to 50 µm. This modification was made for clearer comparison and does not affect the interpretation of the results. Please see figure 7 located on page 26.

Reviewer#1, Comment # 15: Is the optimized synthesis of AuNPs, ranging from 12‒30 nm, for a total flow rate of 5000 μl/min?

Author response: We thank the reviewer for their comment. Within the scope of this work, using the specific helical mixer design and the selected gold precursor and reducing agent concentrations, a total flow rate of 5000 μL/min was identified as the optimized condition for synthesizing gold nanoparticles with an average size of approximately 19 nm, within an overall size range of 14-25 nm.

Reviewer#1, Comment # 16: Can the authors introduce a parameter based on the size and flow rate for a more generic NPs synthesis?

Author response: We thank the reviewer for this valuable suggestion. While introducing a generalized parameter linking nanoparticle size and flow rate is an interesting direction, nanoparticle formation is strongly influenced by system-specific factors such as reaction chemistry, precursor concentration, and nucleation and growth kinetics. As a result, the relationship between flow rate and particle size is not readily universal. In this work, we therefore focus on demonstrating flow rate as an effective tuning parameter for gold nanoparticle synthesis under fixed chemical conditions. Establishing a generalized parameter applicable across different nanoparticle systems would require a broader, systematic study beyond the scope of the present work.________________________________________

Reviewer#1, Comment # 17: Any statistical analysis for the printed chips?

Author response: The primary focus of this work is the development of a hybrid system for nanoparticle fabrication. The 3D-printed helical structure was designed based on established literature to enhance mixing within the microchannel, and the channel dimensions were matched to the helical structure to ensure proper integration. These dimensions were chosen considering the resolution limits of the DLP 3D printer and our previous work, which established conditions suitable for the desired flow rates and nanoparticle synthesis. While statistical analysis of the printed geometry was not performed, we provide detailed statistical analysis of the nanoparticles produced using these chips, which directly reflects the functional performance of the system.

Reviewer#1, Comment # 18: Editorial Proofread for typos. (Typo: neMESYS syringe pump, missing period after “and reduced transparency”)

Author response: We thank the reviewer for the editorial suggestions. The manuscript has been carefully proofread in its entirety. The specific issues noted by the reviewer, including the capitalization of “NeMESYS syringe pump” and the missing period after “and reduced transparency,” have been corrected. In addition, further typographical and formatting issues were addressed throughout the manuscript, including spacing before citations, unit formatting (e.g., µL/min), capitalization consistency, and minor punctuation errors. These revisions were purely editorial and did not affect the scientific content.

Reviewer#2, Comment # 1: Remove the section on the NS equation - this can be referenced as the equation is not required here. We know which equation the solver needs to solve. As this is a case of diffusive mixing, Talor-Aris dispersion should be considered here more so than the fundamental simplified NS equation. However, neither of these is relevant for the manuscript at hand.

Author response: We thank the reviewer for this helpful suggestion. In response, we have removed the exp

---

## [Decision Letter · Decision Letter 1]

29 Jan 2026

Hybrid Passive Micromixer Using Combined Traditional Microfabrication and 3D Printing for Gold Nanoparticle Synthesis

PONE-D-25-52946R1

Dear Dr. Alazzam,

We’re pleased to inform you that your manuscript has been judged scientifically suitable for publication and will be formally accepted for publication once it meets all outstanding technical requirements.

Kind regards,

Bonnie Gray

Academic Editor

PLOS One

Additional Editor Comments (optional):

Reviewers' comments:

Reviewer's Responses to Questions

**Comments to the Author**

Reviewer #1: All comments have been addressed

Reviewer #2: All comments have been addressed

2. Is the manuscript technically sound, and do the data support the conclusions?

Reviewer #1: Yes

Reviewer #2: Yes

3. Has the statistical analysis been performed appropriately and rigorously?

Reviewer #1: Yes

Reviewer #2: Yes

4. Have the authors made all data underlying the findings in their manuscript fully available?

Reviewer #1: Yes

Reviewer #2: Yes

5. Is the manuscript presented in an intelligible fashion and written in standard English?

Reviewer #1: Yes

Reviewer #2: Yes

Reviewer #1: Adding a label on the template printed by Dilase 3D in the soft lithography step of Figure 1 will be more clarifying.

Reviewer #2: The authors have done a good job ironing out the last details and the paper is now ready for publication.

**Do you want your identity to be public for this peer review?** For information about this choice, including consent withdrawal, please see our Privacy Policy

Reviewer #1: No

Reviewer #2: **Yes:** Bastian E. Rapp

---

## [Editor Report · Acceptance letter]

PONE-D-25-52946R1

PLOS One

Dear Dr. Alazzam,

I'm pleased to inform you that your manuscript has been deemed suitable for publication in PLOS One. Congratulations! Your manuscript is now being handed over to our production team.

Kind regards,

on behalf of

Dr. Bonnie Gray

Academic Editor

PLOS One